# The Phenotype of the Adipocytes Derived from Subcutaneous and Visceral ADMSCs Is Altered When They Originate from Morbidly Obese Women: Is There a Memory Effect?

**DOI:** 10.3390/cells11091435

**Published:** 2022-04-23

**Authors:** Agnieszka Mikłosz, Bartłomiej Łukaszuk, Elżbieta Supruniuk, Kamil Grubczak, Aleksandra Starosz, Magdalena Kusaczuk, Monika Naumowicz, Adrian Chabowski

**Affiliations:** 1Department of Physiology, Medical University of Bialystok, Mickiewicza 2C Street, 15-222 Bialystok, Poland; bartlomiej.lukaszuk@umb.edu.pl (B.Ł.); elzbieta.supruniuk@umb.edu.pl (E.S.); adrian.chabowski@umb.edu.pl (A.C.); 2Department of Regenerative Medicine and Immune Regulation, Medical University of Bialystok, Waszyngtona 13 Street, 15-269 Bialystok, Poland; kamil.grubczak@umb.edu.pl (K.G.); aleksandra.starosz@umb.edu.pl (A.S.); 3Department of Pharmaceutical Biochemistry, Medical University of Bialystok, Mickiewicza 2A Street, 15-222 Bialystok, Poland; magdalena.kusaczuk@umb.edu.pl; 4Department of Physical Chemistry, Faculty of Chemistry, University of Bialystok, K. Ciolkowskiego 1K Street, 15-245 Bialystok, Poland; monikan@uwb.edu.pl

**Keywords:** ADMSCs, adipocytes, metabolic syndrome, obesity, subcutaneous and visceral adipose tissue, lipid metabolism

## Abstract

Adipose tissue is an abundant source of mesenchymal stem cells (ADMSCs). Evidence has suggested that depot-specific ADMSCs (obtained from subcutaneous or visceral adipose tissue–subADMSCs or visADMSCs, respectively) account for differential responses of each depot to metabolic challenges. However, little is known about the phenotype and changes in metabolism of the adipocytes derived from ADMSCs of obese individuals. Therefore, we investigated the phenotypic and metabolic characteristics, particularly the lipid profile, of fully differentiated adipocytes derived from ADMSCs of lean and obese (with/without metabolic syndrome) postmenopausal women. We observed a depot-specific pattern, with more pronounced changes present in the adipocytes obtained from subADMSCs. Namely, chronic oversupply of fatty acids (present in morbid obesity) triggered an increase in CD36/SR-B2 and FATP4 protein content (total and cell surface), which translated to an increased LCFA influx (^3^H-palmitate uptake). This was associated with the accumulation of TAG and DAG in these cells. Furthermore, we observed that the adipocytes of visADMSCs origin were larger and showed smaller granularity than their counterparts of subADMSCs descent. Although ADMSCs were cultured in vitro, in a fatty acids-deprived environment, obesity significantly influenced the functionality of the progenitor adipocytes, suggesting the existence of a memory effect.

## 1. Introduction

Over the past few decades, the prevalence of obesity has increased dramatically, and its consequences represent a major public health concern. A chronic imbalance between energy intake and energy expenditure leads to adipose tissue (AT) dysfunction, including adipocyte hypertrophy, disequilibrium between lipogenesis and lipolysis, abnormal secretion of numerous adipokines and cytokines, and infiltration of fat tissue with inflammatory M1 macrophages [1]. The dysfunctional AT is unable to efficiently process the circulating metabolites, which increases the risk of metabolic disorders. White adipose tissue is a highly heterogeneous organ distributed throughout the body, but its principal depots are the subcutaneous adipose tissue (SAT) and visceral adipose tissue (VAT) [2]. Although morphologically similar, SAT and VAT show distinct characteristics, they differ in terms of gene expression pattern, endocrine and metabolic activity, insulin sensitivity, vascularization, innervation and infiltration by immune cells [3,4]. This functional heterogeneity and diverse anatomical distribution translate into their divergent involvement in the formation of metabolic disorders. In fact, VAT expansion is a major predictive factor for the development of metabolic abnormalities, such as insulin resistance (IR), lipodystrophy and type 2 diabetes mellitus (T2DM), whereas the enlargement of subcutaneous fat improves insulin sensitivity, reduces metabolic complications, and therefore, is considered to be a protective [5]. However, in obesity, superfluous energy supply exceeds the storage capacity of SAT, resulting in lipid deposition in VAT and non-adipose tissues (ectopic lipid accumulation). Therefore, the comparison of adipose depots with respect to their distinct anatomic location is crucial to clarify their role in obesity-related complications.

Postmenopausal women are one of the groups most prone to develop metabolic disorders as a result of the loss of endogenous sex hormone production. Menopause changes the phenotype of adipose tissue causing adipocyte hypertrophy, which translates to tissue hypoxia, triggering its inflammation, and fibrosis. Moreover, inadequate estrogen-mediated ability to store excess lipids in SAT may contribute to their deposition in less favourable anatomic locations [6,7]. Still the protective role of SAT against the negative effects of obesity is highly debated [8].

Adipose-derived mesenchymal stem cells (ADMSCs) are a new source of stem cells that can be easily isolated from adipose tissue [1]. Importantly, ADMSCs of subcutaneous or visceral ancestry exhibit morphological and molecular differences. Subcutaneous ADMSCs are characterized by a higher proliferation rate and greater potential towards adipogenic differentiation, moreover, they display an increased capacity to store lipids and pronounced ability for adiponectin secretion compared with visceral ADMSCs [9,10]. In addition to the inherent properties of ADMSCs [11,12], also the microenvironment surrounding the cells [12,13] may underlie the biological functions of adipose depots. In response to excessive caloric intake, pluripotent ADMSCs undergo transformation to preadipocytes; thus, ADMSCs constitute an almost unlimited source of these cells [14]. However, obesity disrupts ADMSCs functioning by reducing their lipid storage ability, compromising their regenerative potential, modifying their apoptosis susceptibility and endocrine/paracrine function (e.g., by stimulating cytokine secretion) [15,16,17,18]. For example, visceral ADMSCs in obese humans secrete more pro-inflammatory cytokines compared to subcutaneous ADMSCs, which is consistent with the stronger pro-inflammatory pattern adopted by VAT during the development of the metabolic dysfunction [16,19]. Still, the distinct effects of nutrient overflow on the phenotype of ADMSCs have only recently been started to be unravelled.

Because, the differences between the particular white adipose tissue depots may well come from the ADMSCs properties, we decided to thoroughly investigate the intrinsic characteristics of the adipocytes differentiated from subcutaneous and visceral ADMSCs. Knowing that adipocytes’ functionality may be lost during obesity, we obtained the cells (ADMSCs) from lean as well as morbidly obese (with and without metabolic syndrome) postmenopausal female subjects and differentiated them into mature adipocytes. Particularly, we determined the effects of obesity and metabolic syndrome on the expression and cellular localization of several protein carriers responsible for the uptake of long chain fatty acids (LCFAs). Furthermore, we examined the expression of proteins involved in lipid turnover, in conjunction with the cellular lipid profile in the adipocytes differentiated from the ADMSCs.

## 2. Materials and Methods

### 2.1. The Origin of Human Adipose-Derived Mesenchymal Stem Cells (ADMSCs)

Subcutaneous (abdominal region) and visceral (omental region) white adipose tissue biopsies (approximately 2–4 g) were obtained from postmenopausal female subjects treated at the First Department of General and Endocrine Surgery at the University Hospital in Białystok. The study protocol was approved by the Ethics Committee of the Medical University of Bialystok (permission R-I-002/187/2017), in agreement with the Declaration of Helsinki. All patients were informed about the properties of the study and gave their written consent before participating in the study. Enrolled patients underwent clinical examination, anthropometric measurements and appropriate laboratory tests (Appendix A). Major exclusion criteria were acute inflammatory process and a history of malignancy. The lean group consisted of four women of normal BMI value (19–24.9 kg/m^2^) who underwent elective laparoscopic cholecystectomy. Patients with obesity were divided into morbidly obese female individuals of BMI > 40 kg/m^2^ without metabolic syndrome (*n* = 4) and with metabolic syndrome (*n* = 4) who underwent sleeve gastrectomy. Immediately following dissection, the samples were placed in phosphate-buffered saline (PBS, PAN-Biotech, Aidenbach, Germany) and transported to the laboratory for ADMSCs isolation.

Based on the available literature we have developed an isolation protocol for ADMSCs as described previously [20]. Briefly, adipose tissue samples were washed in PBS, cleaned from visible blood vessels and then minced and digested in collagenase (250 U/mL collagenase NB 4G Proved Grade, Serva, Heidelberg, Germany) at 37 °C for approximately 1 h, until complete digestion. Mature adipocytes from the stromal vascular fraction were removed using a 500 µm strainer. After centrifuging, the cell pellet was resuspended in erythrocyte lysis buffer (Thermo Fisher Scientific, Waltham, MA, USA), filtered through a 200 µm and then 20 µm strainers and centrifuged again for 5 min at 600× *g*. The ADMSCs-rich pellet was suspended in mesenchymal stem cells medium containing growth supplements (MSCM, ScienCell Research Laboratories, Carlsbad, CA, USA), 5% fetal bovine serum (FBS, Thermo Fisher Scientific, Waltham, MA, USA) and antibiotics (PAN-Biotech, Aidenbach, Germany). The viability and cell purity of isolated ADMSCs was determined via by flow cytometry as described below. The culture medium was changed every 2–3 days until cells reached the confluency of 80–90%; then they were detached with trypLE (Thermo Fisher Scientific, Waltham, MA, USA). ADMSCs were then collected at a density of 5 × 10^5^ − 5 × 10^6^ cells in stem cell cryopreservation medium (Stem-Cell banker DMSO Free, Takara Bio, Mountain View, CA, USA) and stored at −80 °C.

### 2.2. ADMSCs Differentiation towards Adipocytes

#### 2.2.1. Adipogenesis of ADMSCs

For further experiments, cells from 2 to 4 passages were thawed and seeded in MSCM on the appropriate plates. Once the cell confluency reached approximately 90%, adipogenesis was induced by a differentiation medium containing supplements (MADM, Mesenchymal Stem Cell Adipogenic Differentiation Medium, ScienCell Research Laboratories, Carlsbad, CA, USA), 5% FBS and 1% penicillin/streptomycin solution. The medium was replaced every 3–4 days, and the progress of adipogenesis was monitored by microscopic observation of lipid vacuoles in the cells. After 14–21 days, the adipocytes were harvested and subjected to specific analyses.

#### 2.2.2. Assessment of the Accumulated Lipids

The amount of accumulated lipids was measured in mature adipocytes after fixing with 10% formalin (Sigma Aldrich, St. Louis, MO, USA) for 30 min at RT. Then, the cells were incubated with 0.5% Oil Red-O solution (Sigma Aldrich) for 1 h, and the overstaining was washed several times with PBS. The images were created using an inverted microscope (Olympus, magnification ×400). Next, the accumulated Oil Red-O was extracted with 100% isopropanol and the absorbance was measured at 510 nm using a microplate reader (Synergy H1 Hybrid Reader, BioTek, Santa Clara, CA, USA). Undifferentiated ADMSCs stained with Oil Red-O served as control. Oil Red-O concentration was determined based on known standards of Oil Red-O (0.02 mg/mL to 2.5 mg/mL).

Additionally, the average size and the number of lipid droplets per cell were evaluated based on the microphotographs using OLYMPUS cellSens Standard 1.18 software.

### 2.3. Flow Cytometry Characterization

Adipocytes collected after cell culture were subjected to immunostaining with monoclonal antibodies: anti-FATP1 (ACSVL5, mouse anti-human) (R&D Systems, Minneapolis, MN, USA); anti-FATP4 (ACSVL4, rabbit anti-human); anti-FABP4 (A-FABP, rabbit anti-human); and anti-CD36 (SR-B2; rabbit anti-human) (Abcam, Cambridge, UK). It should be noted that prior to staining procedures, the adipocytes were divided into two separate groups, one for extracellular staining only, and another permeablized with FACS Permeabilizing Solution 2 (BD Bioscience, Franklin Lakes, NJ, USA) for additional detection of intracellular markers. Approximately 100,547 of the collected cells were used, the number of cells recovered was 45,689 for extracellular events and 15,105 for intracellular events. Following incubation at room temperature, in dark conditions, cells were washed in phosphate-buffered saline (PBS with no calcium and magnesium; Corning, Corning, NY, USA). Subsequently, secondary detection antibodies were used to bind to antibodies bound to adipocytes proteins: goat anti-rabbit (Alexa Fluor 488); and goat anti-mouse (Alexa Fluor 647) (Invitrogen, MA, USA). Incubation was followed by double centrifugation in PBS to wash out unbound antibodies. Finally, cells were fixed using CellFIX (BD Biosciences) and stored in 4 °C until acquisition using a FACS Calibur flow cytometer (BD Biosciences; San Jose, CA, USA). Processing of the flow cytometric data was performed with the use of FlowJo software (TreeStar Inc., Ashland, OR, USA). The studied proteins were analyzed within adipocytes distinguished initially on the basis of morphological properties–forward scatter (FSC; relative size) and side scatter (SSC; relative granularity/complexity) (Appendix A). Lack of 7AAD-related fluorescence was used to gate viable cells. The obtained results were presented as frequencies of selected markers within adipocytes or mean fluorescence intensity (MFI) of a specific marker within adipocytes. For the assessment of relative changes in size and granularity/complexity, adipocytes were divided into four separate subgroups with increasing values of each morphological parameter.

### 2.4. RNA Isolation and Quantitative Real Time RT-PCR

Total cellular RNA was extracted using TRIzol Reagent (Sigma Aldrich, Saint Louis, MO, USA) in accordance with the manufacturer’s instructions. RNA concentration and purity were assessed by spectrophotometry (at an absorbance OD ratio of 260/280 and 260/230). Reverse transcription was performed using the EvoScript universal cDNA master kit (Roche Molecular Systems, Boston, MA, USA) with 1 µg of total RNA. Quantitative real time polymerase chain reaction (qRT-PCR) was carried out in duplicate using the LightCycler 96 System Real-Time thermal cycler with FastStart essential DNA green master (Roche Molecular Systems) as the detection dye. The following reaction parameters were applied in a thermal cycler: 15 s denaturation at 94 °C, 15 s annealing at 57 °C for RPLO13A and TBC1D1, 58 °C for TBC1D4 and CD36/SR-B2, 59 °C for FATP1 and FATP4, and 62 °C for FABPpm, and then a 15 s extension at 72 °C for 45 cycles. The primer sequences are listed in Appendix A. The primers’ efficiency was analysed using the standard curve method. Gene expression was calculated according to the Pfaffl method [21], normalizing to the housekeeper gene (RPLO13A).

### 2.5. Immunoblotting

Routine Western blotting procedures were used to detect protein content in total lysate [20]. In brief, ADMSCs lysates were prepared using ice-cold radioimmunoprecipitation assay (RIPA) buffer containing a mix of protease and phosphatase inhibitors (Roche Diagnostics GmbH, Mannheim, Germany). The total protein concentration was assayed using the BCA method with bovine serum albumin (BSA) as a standard. Then, lysates were reconstituted in Laemmli buffer (Bio-Rad, Hercules, CA, USA), and equal amounts of the proteins (30 µg per sample) were loaded on Criterion TGX Stain-Free Precast Gels (Bio-Rad, Hercules, CA, USA) for sodium dodecyl-sulfate polyacrylamide gel electrophoresis (SDS-PAGE). Size-separated proteins were transferred onto polyvinylidene difluoride (PVDF) membranes. After blocking in 5% non-fat dry milk for 1 h, membranes were incubated overnight at 4 °C with the corresponding primary antibodies, i.e., TBC1D1 (1:500, cat. no. H00023216-B02P, Novus Biologicals, CO, USA), AS160/TBC1D4 (1:1000, cat. no. # 07-741, Merck Millipore, CA, USA), CD36/SR-B2 (1:500, cat. no. sc-7309, Santa Cruz Biotechnology, Inc., Dallas, TX, USA), FATP4 (1:1000, cat. no. ab200353, Abcam, Cambridge, UK), FATP1 (1:500, cat. no. sc-25541, Santa Cruz Biotechnology, Inc., Dallas, TX, USA), FABPpm (1:4000, ab180162, Abcam, Cambridge, UK), β-HAD (1:1000, cat. no. sc-271495, Santa Cruz Biotechnology, Inc., Dallas, TX, USA), DGAT1 (1:500, cat. no. sc-271934, Santa Cruz Biotechnology, Inc., Dallas, TX, USA), ATGL (1:500, cat. no. sc-365278, Santa Cruz Biotechnology, Inc., Dallas, TX, USA), FASN (1:1000, cat. no. ab128870, Abcam, Cambridge, UK), GAPDH (1:1000, sc-47724, Santa Cruz Biotechnology, Inc., Dallas, TX, USA). Thereafter, bound antibodies were detected with suitable anti-rabbit or anti-goat IgG horseradish peroxidase-conjugate secondary antibodies (1:3000, Santa Cruz Biotechnology, Dallas, TX, USA). The protein bands were imaged by chemiluminescence using Clarity Western ECL Substrate (Bio-Rad, Hercules, CA, USA), and signal intensities were quantified densitometrically using a ChemiDoc visualisation system (Bio-Rad, Hercules, CA, USA). The protein expression (optical density arbitrary units) was normalised to GAPDH expressions and was related to the control subADMSCs group.

### 2.6. 9,10-[^3^H]-Palmitic Acid Uptake

Before the analysis, differentiated adipocytes were starved in a serum-free medium for 3 h. Then the cells were incubated with Krebs–Ringer-HEPES buffer supplemented with palmitic acid (Sigma Aldrich, St. Louis, MO, USA) bound to fatty acids-free bovine serum albumin (BSA, Sigma Aldrich) with the radiolabelled 9,10-[^3^H] palmitic acid (Perkin Elmer) at the specific activity of 1 μCi mL for 5 min at 37 °C/5% CO_2_. Afterwards, the reaction was terminated by addition of ice-cold PBS and finally the cells were solubilised in 0.05 N NaOH. Radioactivity was measured using a Packard TRI-CARB 1900 TR scintillation counter, and was normalised to the protein concentrations.

### 2.7. Lipid Content Quantification (Gas Liquid Chromatography)

Lipids from the ADMSCs were extracted using chloroform-methanol solution according to the Folch method and separated into different fractions using thin-layer chromatography (TLC). Individual fatty acids (FAs) from each fraction were methylated and then esters were quantitatively determined in relation to corresponding retention times of standards by means of GLC method (Hawlett-Packard 5890 Series II gas chromatograph, HP-INNOWax capillary column). The concentrations of FFA, DAG and TAG were assessed as the sum of the individual FAs in each fraction.

### 2.8. Statistical Analysis

The numbers of patients (n–number of patients per group) included in the analysis are mentioned in the legends of the appropriate Figures/Tables. The measurements were made in triplicate or duplicate (see Figure legends) and the arithmetic means were used for subsequent investigation. Statistical analyses were performed with R (ver. 3.6.3) or GraphPad Prism (9.0.0) programs. The Shapiro–Wilk test (test for normality) and Fligner–Killeen tests (test for homogeneity of variances) were used to determine the subsequent application of parametric or non-parametric methods. The data were analysed using three-way ANOVA (Appendix A). In addition to the above, the data were analysed with either the Student’s *t*-test, or Wilcoxon rank sum test.

## 3. Results

### 3.1. Characterization of Human ADMSCs

First, using specific markers, we characterized the ADMSCs immunophenotype by flow cytometry. As previously reported by us [20], the ADMSCs showed high expressions of CD105, CD73, and CD90 (the proportion of positive cells exceeded 99%) and lacked the expression of CD45 and lineage markers; negative markers were expressed in less than 1% of the cells. Additionally, high expression of CD10 and low expression of CD105 distinguished subADMSCs from visADMSCs. Thereafter, the differentiation into mesoderm derivatives, i.e., osteocytes, chondrocytes, and adipocytes, confirmed the multilineage potential of the isolated cells. The effectiveness of the above was confirmed upon microscopic inspection and by immunocytochemistry [20]. Moreover, neither obesity nor the presence of the metabolic syndrome affects the viability of ADMSCs derived from both subcutaneous and visceral adipose tissues (Figure 1A). To confirm adipogenesis in stem cells obtained from patient donors with different metabolic status, the cells were stained with Oil Red O (ORO) for the identification of lipid droplets. Once differentiated, all the cells presented a large number of lipid-filled vacuoles that appeared red under an inverted light microscope. Quantitative analysis of ORO demonstrated a pronounced build-up of intracellular lipids in the adipocytes when compared with undifferentiated precursor cells (Figure 1B). The average cell size was relatively stable in the adipocytes of subADMSC ancestry (Figure 1C). In the cells originating from visceral tissue, we observed greater cell areas in the cells from visObese(-) and visObese(+) groups when compared with visLean. We observed no statistically significant changes between the analysed groups with respect to the number of lipid droplets per cell (Figure 1C). The mean average cell size was unchanged in the adipocytes Further, based on the results obtained from flow cytometry analysis, FABP4 mean fluorescence intensity (MFI) was diminished in adipocytes derived from subADMSCs isolated from obese women with metabolic syndrome, when compared to the lean control group. However, this does not apply to adipocytes originating from visADMSCs. Moreover, the adipocytes obtained from the visADMSCs of the lean patients appeared to have a lower expression of FABP4 compared to their subcutaneous counterparts (Figure 1D).

### 3.2. SLC27A4/FATP4 and CD36/SR-B2, but Not FATP1 or FABPpm, Contribute to Increased Fatty Acid Uptake in Obese Adipocytes Derived from subADMSCs

The rate of cellular fatty acid (FA) uptake is short- and long-term regulated. Short-term regulation (i.e., minutes) occurs via reversible recycling of CD36/SR-B2 or FATP4 from intracellular compartments to the plasma membrane, which is dependent on Rab-GTPase activating proteins (RabGAPs), namely TBC1D1 and/or TBC1D4 (AS160) [22]. As we detected virtually no changes in the amount of TBC1D1 at both mRNA and protein levels, it seems that only AS160 had a regulatory role in the adipocytes (Figure 2A–C). An examination of TBC1D4 levels revealed its greater expression in the adipocytes derived from the obese patients (with and without metabolic syndrome) when compared to their lean analogues. Herein, the mRNA and protein level of AS160 was increased by roughly 40% and 60% in the case of subcutaneous tissue. In visADMSCs, the protein expression of TBC1D4 was also markedly elevated (+76% and 79% for visObese(-) and visObese(+) vs. visLean, *p* < 0.05). In addition to the above-mentioned alterations, we observed a lower TBC1D4 protein content in the cells of visceral origin when compared to their subcutaneous counterparts (Figure 2A–C).

Long-term regulation of the rate of cellular FA uptake, which occurs in obesity (high fatty acid supply), involves changes in the gene transcription and/or protein abundance. In our study, CD36/SR-B2 mRNA was not altered in obese adipocytes compared with lean cells derived from both sub- and visADMSCs. Interestingly, the presence of metabolic syndrome significantly augmented the transcript level of this transporter (+74% and +87% for subObese(+) vs. subObese(-) and visObese(+) vs. visObese(-), *p* < 0.05, Figure 3A). However, the protein abundance was solely increased in subADMSCs mature adipocytes that originated from obese but metabolically healthy patients compared to the lean control group (+59%), whereas both obese groups exhibited higher CD36/SR-B2 levels in the adipocytes derived from visADMSC (+117% and +166 % for visObese(-) and visObese(+) vs visLean, respectively, *p* < 0.05, Figure 3B). Furthermore, we observed no changes in the expression of FABPpm (neither at mRNA nor protein level) with respect to any of the investigated groups (Figure 3A,B). Similarly, neither mRNA nor the total protein expression of FATP1 were significantly altered in any of the examined groups in the adipocytes derived from subADMSCs (Figure 3A,B). On the contrary, the adipocytes of visceral provenance displayed a characteristic pattern, i.e., the fat cells derived from ADMSCs of obese patients had a significantly lower total FATP1 protein content when compared with their analogues stemming from the lean patients (*p* < 0.05). The examination of FATP4 mRNA and total protein content revealed changes, but only in the adipocytes differentiated from subADMSCs. We observed a greater FATP4 transcript content in the case of the cells derived from the obese patients (with and without metabolic syndrome) when compared with their counterparts descending from the lean individuals (+73% and +50% for subObese(-) and subObese(+) vs. subLean, *p* < 0.05, Figure 3A). The above-mentioned pattern was also apparent in the total protein content; the cells derived from obese women had a significantly greater expression of FATP4 compared to the subLean group (+115% and +103% for subObese(-) and subObese(+) vs. subLean, *p* < 0.05, Figure 3B). Moreover, we observed some differences in the protein content with respect to the cells’ tissue of origin. In general, the adipocytes descending from visADMSCs showed lower CD36/SR-B2 and FATP4 protein content than their counterparts derived from subADMSCs (Figure 3B). 

To examine the changes in subcellular localization of FA transporters, we performed flow cytometry analysis. The surface expression of the two most abundant FA transporters in adipocytes, i.e., CD36/SR-B2 and FATP4 was significantly higher in the cells differentiated from subADMSCs of obese, metabolically unhealthy women (Figure 4). It may suggest that both the transporters under high fatty acid supply (as in obesity) are permanently relocated to the adipocytes surface, increasing FA uptake and further progressive lipid accumulation. In contrast, obesity itself did not change the subcellular location of FATP1 in the adipocytes derived from both adipose tissues.

The rate of long-chain fatty acid influx was determined by ^3^H-palmitate uptake. In line with the results described above, the adipocytes originated from subADMSCs of obese patients with metabolic syndrome showed a 145% increase in ^3^H-palmitate uptake, when compared with the adipocytes from lean ADMSCs (Figure 2D). Likewise, palmitate uptake was increased in the adipocytes stemming from obese patients with metabolic syndrome of visceral ancestry. However, the above-mentioned changes did not reach the level of statistical significance (*p* > 0.05). In addition to the above, we observed a significantly lower palmitate uptake while comparing visceral and subcutaneous cells derived from obese patients with metabolic syndrome (−44%, *p* < 0.05, Figure 2D).

### 3.3. Effect of ADMSCs Tissue Origin and the Metabolic Status of the Patient Donor on Adipocyte Size and Granularity

Using the flow cytometry analysis (forward scatter (FSC) = positive correlation with the size of cells) we found that the adipocytes differentiated from visADMSCs were larger than those from subADMSCs (Figure 5A). The size of mature adipocytes and the size distribution were significantly different between morbidly obese and lean postmenopausal women with respect to visADMSCs-derived cells. For example, the adipocytes from the visObese(+) group had greater cell volume compared to their visLean and visObese(-) counterparts. In addition, a higher incidence of intermediate adipocyte sizes was noted for ranges II (275–550) and III (550–825) (Figure 5A). On the other hand, the metabolic status of the donor patient did not affect the cell size or the size distribution in the adipocytes differentiated from subADMSCs. Furthermore, analysis of the side scatter (SSC) data allowed for the estimation of adipocyte granularity. The cells of visceral origin showed less granularity, indicating their tendency to accumulate lipids in several spacious vacuoles. In all the studied groups, the granularity of the adipocytes appeared to decrease with the size of the cells (Figure 5B).

### 3.4. Enhanced FA Uptake Promotes Lipid Accumulation Together with Changes in the Composition of Different FA Species Mainly in the Adipocytes Derived from subADMSCs of Morbidly Obese Women with Metabolic Syndrome

The total fatty acid content of FFA was increased only in the adipocytes differentiated from subADMSCs of obese patients with metabolic syndrome (+45% subObese(+) vs. subLean, *p* < 0.05, Figure 6A). This was mostly caused by an increase in the concentration of saturated fatty acid species, i.e., palmitic acid (C16:0), while no changes in unsaturated FA levels were observed. When comparing the adipocytes of different tissue descent we observed a lower total FFA content in the cells derived from visADMSCs, but only in the case of obese patients with metabolic syndrome (−66% for visObese(+) vs. subObese(+), *p* < 0.05, Figure 6A). This was due to their lower saturated (C16:0), and also unsaturated fatty acids content (UNSFA), i.e., palmitooleic (C16:1), arachidonic (20:4n6) (Appendix A).

The adipocytes derived from the ADMSCs of obese patients with metabolic syndrome also had a greater total DAG concentration when compared with the cells of other patient type descent (Figure 6B). However, this was evident only in the case of the adipocytes of subcutaneous origin (+66% for subObese(+) vs. subLean, *p* < 0.05; and +37% for subObese(+) vs. subObese(-), *p* < 0.05). Tissue comparison revealed that the adipocytes stemming from the visADMSCs of obese patients with metabolic syndrome had a lower total DAG level compared with their counterparts derived from the subADMSCs (−47% for visObese(+) vs. subObese(+), *p* < 0.05). This was most likely caused by a decrease in the amount of unsaturated, i.e., palmitooleic (C16:1) and oleic (18:1n9c), fatty acids content (Appendix A).

Similarly to the total FFA and DAG levels, TAG content in the cells differentiated from subADMSCs was significantly increased in the case of patients with metabolic syndrome in comparison with the lean adipocytes (+30% for subObese(+) vs. subLean, *p* < 0.05, Figure 6C). This was probably due to a greater concentration of saturated fatty acid species with unchanged UNSFA in this group. Moreover, we did not observe any statistically significant changes in the total amount of TAG in the cells of visceral origin with respect to the patient type. However, a comparison of the adipocytes derived from the sub- and visADMSCs revealed a lower total concentration of TAG in the cells of visceral provenance (−68% and −74% for visLean vs. subLean and for visObese(+) vs. subObese(+), respectively, *p* < 0.05). A more detailed analysis of the individual content of distinct FA species within the adipocytes revealed lower amounts of saturated fatty acids such as palmitic acid (C16:0) as well as unsaturated fatty acids content, i.e., oleic acid (18:1n9c), linoleic acid (18:2n6c) and linolenic acid (C18:9n3) (Appendix A).

### 3.5. LCFAs Synthesis and Utilization Are Enhanced in the Adipocytes Derived from ADMSCs of Morbidly Obese Women

The expression of fatty acid synthase (FASN), an enzyme involved in de novo lipogenesis as well as diacylglycerol acyltransferase (DGAT1) and which catalyzes the final step in the biosynthesis of TAGs, was markedly elevated in the adipocytes derived from both sub- and visADMSCs of obese patients (with or without metabolic syndrome). In the case of the cells of subcutaneous origin, the amount of FASN was greater by 81% and 97%, whereas DGAT1 expression was increased by 84% and 112% (for subObese(-) and subObese(+) vs. subLean, *p* < 0.05, Figure 7). The above-mentioned pattern was also visible in the cells that descended from visADMSCs (FASN: +132% and +105%; DGAT1: +180% and +302%, for visObese(-) and visObese(+) vs. visLean, *p* < 0.05). In addition to the above-mentioned changes, we observed that FASN and DGAT1 expression was lower in the cells that descend from visceral fat depots in comparison to their counterparts of subcutaneous origin.

Western Blot analysis of protein expression of beta-hydroxyacyl CoA dehydrogenase (β-HAD) revealed that the adipocytes derived from subADMSCs had an increased amount of β-HAD in the Obese(+) group, an even greater change was observed in the cells from the Obese(-) group, compared with the lean controls (+36% and +108% for subObese(+) and subObese(-) vs. subLean, *p* < 0.05, Figure 7). The cells of visceral provenance in the obese group without metabolic syndrome also showed the greatest expression of β-HAD protein when compared with the lean controls (+47% for visObese(-) vs. visLean, *p* < 0.05).

The level of adipose triglyceride lipase (ATGL) in the adipocytes derived from subADMSCs was significantly increased solely in the Obese(-) group in comparison with the control group (Figure 7). Surprisingly, in the cells of visADMSCs provenance we observed a decreased amount of ATGL protein in the obese patients compared with the lean subjects. We also observed a difference in the amount of ATGL between the cells of subcutaneous and visceral origin. The latter had a lower cellular level of ATGL, which was especially visible in the case of obese patients (−83% for visObese(-) vs. subObese(-), *p* < 0.05; and −90% for visObese(+) vs. subObese(+), *p* < 0.05).

## 4. Discussion

A growing body of evidence points to the existence of metabolic and functional differences between individual fat depots. In consequence, ADMSCs obtained from different locations vary with respect to their morphology, function and biochemical/metabolic properties, as well as gene expression patterns. These characteristics are stable and are maintained after the ADMSCs have been isolated and cultured in vitro [23]. Although the role of the cells constituting white adipocyte tissue (WAT) has been well recognized in obesity, the metabolism of adipocytes differentiated from human mesenchymal stem cells is still being elucidated. Importantly, WAT is a highly heterogeneous tissue in which mature adipocytes account for only 15–30% of the total adipose cell fraction, the rest is stromal vascular fraction (SVF) [24,25]. Thus, a study of the adipocytes derived from the tissue stem cells should allow the observation of ‘pure’ (i.e., not obfuscated by other cellular fractions, nor the tissue milieu) adipocyte phenotypes. Here, we decided to compare the makeup and lipid profile of the adipocytes originated from ADMSCs of lean and morbidly obese women (with or without metabolic syndrome) to better understand intrinsic factors behind each condition.

In adipose tissue, the cellular uptake of LCFAs is facilitated by several membrane-associated proteins, including CD36/SR-B2, FABPpm and FA transport proteins (FATP1 and FATP4) [26]. Our data show distinct expression profiles of fatty acid handling proteins in the adipocytes of sub- and visADMSCs origin. The levels of two of the most abundant FA transporters, CD36/SR-B2 and FATP4, were distinctly greater in the adipocytes differentiated from subADMSCs than in those from visADMSCs. These results may imply different fatty acid handling in various adipose tissue depots. Previously, it was suggested that both CD36/SR-B2 and FATP4 are important fatty acids transporters, especially in the case of low extracellular FAs concentration [27,28]. This could be the case in our in vitro experiment (low fatty acid concentration in medium). In line with that notion, we observed a higher total TAG content in the subLean group in comparison with their counterparts from the visLean group. However, it is not an obvious effect, since palmitate uptake was similar in both groups. Obesity was associated with greater protein expression of CD36/SR-B2 in the adipocytes originated from ADMSCs of both depots and FATP4 (only in ADMSCs derived from subcutaneous depots). This is consistent with the higher CD36/SR-B2 level observed in vivo by Bonen et al. for VAT and SAT of obese individuals [29], and indicates the fundamental role of subcutaneous adipose tissue in the FA turnover. Our finding is also supported by the results from the examination of monozygotic twins provided by Gertow and co-workers [30]. The authors reported moderate to strong positive correlation between the amount of mRNA for the transporters and the volume of subcutaneous fat (+0.62 for FATP4, and +0.71 for CD36/SR-B2). Interestingly, only a weak correlation (or no correlation at all) was noted for the transporters and intra-abdominal (visceral) fat size (+0.3 for FATP4, and +0.14 for CD36/SR-B2) [30]. Moreover, the adipocytes from the subObese(+) group are characterized by the redistribution of CD36/SR-B2 and FATP4 towards the plasma membrane, as evidenced by the flow cytometry. This translates into the pronounced increased influx of the LCFA (radio-isotope labeled palmitic acid uptake) observed in the subObese(+) group (Figure 8). Such a lipid oversupply is believed to trigger the metabolic complications observed in obese patients with metabolic syndrome. Recently, we have demonstrated that AS160/TBC1D4 and its structural homolog TBC1D1 could be involved in the cellular redistribution of fatty acid transporters. Silencing of TBC1D4, but not TBC1D1, in the adipocytes led to a greater translocation of fatty acid transporters (mostly CD36/SR-B2) into the plasma membrane [20]. Interestingly, in the current study we observed an increased expression of AS160/TBC1D4 in the adipocytes derived from ADMSCs of obese patients (especially from individuals with metabolic syndrome). This should translate into lower plasmalemmal expression of fatty acid transporters and smaller LCFA uptake by the cells, since AS160/TBC1D4 serves as a negative regulator for the translocation of transporters to the cell surface [31]. However, this prediction is in contrast to the actual observations mentioned above (specifically the data from flow cytometry for CD36/SR-B2, FATP4, and palmitic acid uptake). The discrepancy is perplexing and not straightforward to explain. Still, recent review papers by Glatz et al. may shed some light on the topic [22,32]. The authors point to the existence of two mechanisms regulating CD36/SR-B2 mediated FA uptake in skeletal muscle and cardiomyocytes. Long-term regulation is triggered by high fatty acids oversupply (such as occurs in obese patients in vivo) and results in increased CD36/SR-B2 gene transcription. Short-term regulation, on the other hand, is initiated by low fatty acids supply (as observed in vitro) and relies on CD36/SR-B2 subcellular recycling [22,32]. We postulate, therefore, that adipocytes from the obese group have an activated long-term overexpression of fatty acid transporters (CD36/SR-B2, FATP4 at mRNA and protein level) that cannot be exactly counterbalanced by their decreased plasmalemmal translocation (by TBC1D4 overexpression). The net result of this interplay of factors is an increased amount of FA transporters on the cell surface and propensity towards greater fatty acids intake (larger palmitate uptake). On the other hand, neither FABPpm nor FATP1 total protein content was changed in subADMSCs, whereas FATP1 expression was even diminished in the adipocytes from visADMSCs. These results, along with the fact that mRNA expression of FATP1 is relatively low in human adipose tissue, suggest that this transporter does not play a crucial role in LCFAs uptake in obese patients. In line with this notion, Binnert et al. found that the transcript level of FATP1 was not correlated with BMI in male subjects’ subcutaneous adipose tissue [33]. Nevertheless, there are very few reports concerning the fatty acid transporters in human adipocytes differentiated from ADMSCs, and further investigations are required to directly confirm the above-mentioned suppositions.

Adipocytes primarily store and release free fatty acids (FFA) to support local and systemic metabolic demands. The majority, i.e., approximately 80%, of body fat is deposited in the subcutaneous area where the excess of FFA and glycerol is stockpiled as TAG. On the other hand, visceral fat accounts for about 10–20% of total body fat in lean men and less than 10% in healthy women [34]. Studies have shown that subcutaneous adipose tissue increases in volume by hyperplasia rather than by hypertrophy. The opposite is true for visceral fat [35]. This tendency was also maintained in our study. Overall, the adipocytes differentiated from visADMSCs were larger than those of subADMSCs provenance. Considering the metabolic status of the donor patient, obesity significantly increased cell volumes, but only in the adipocytes derived from visADMSCs. Further cytometric analyses demonstrated smaller granularity of the cells of visceral origin, which points to their propensity to accumulate lipids in a few spacious vacuoles. This agrees also with the evaluation using micro-photographs of the average cell size and number of lipid droplets per cell (greater cell size of the adipocytes in visObese(-) and visObese(+) with the unchanged number of lipid droplets suggests an increased size of the latter). Nevertheless, the larger adipocytes in the visceral depot have a decreased capacity to take up an excess of FFAs compared with the new smaller adipocytes differentiated from subADMSCs. This notion is supported by an increased TAG accumulation in the fully differentiated adipocytes of subcutaneous ancestry. Kim et at. reported that subcutaneous ADMSCs had a higher capacity to proliferate and differentiate into adipogenic lineages than those from visceral ADMSCs [36]. Accordingly, in the present study, we found a higher expression of FABP4, an adipogenic marker, in adipocytes derived from lean subADMSCs, which was confirmed in the study by Baglioni and co-workers [11].

The adipocytes of subcutaneous origin (subADMSCs) seem to better reflect the metabolic status of donor patients compared with visceral-derived ADMSCs, since we found that subObese(+) had more lipids (TAG, DAG, and FFA) in comparison with subLean (Figure 8). This is quite interesting because, in general, visceral fat is considered to be more (patho)physiologically active with respect to the induction of cardiovascular and metabolic complications of obesity [5]. However, the progression of the disorder runs rather in the opposite direction (first subcutaneous then visceral fat accumulation and malfunction). In overweight individuals, superfluous lipids are deposited mostly in subcutaneous tissue [24,37]. With progress towards obesity and metabolic syndrome development the lipids are shifted towards visceral storage sites. At some point, hypertrophic adipocytes in the visceral locations approach the limit of their storage capacity. Thus, further overflow of VAT with FFAs results in their ectopic deposition in adjacent organs, e.g., in the liver, advancing the pathological condition even farther [24]. The expression of enzymes involved in lipid synthesis is also in line with that notion. The examined proteins (FASN and DGAT1) presented clearly greater levels in the adipocytes obtained from subADMSCs than those from visADMSCs, which may underlie the higher lipid-storing capacity of the cells of subcutaneous deposits (Figure 8). This finding is supported by previous results of other authors, in which the genes engaged in lipogenesis, such as FASN, acetyl-CoA carboxylase α (ACACA), and stearoyl-CoA desaturase (SCD1), were upregulated in differentiated adipocytes derived from SAT of lean individuals [38]. While the concomitant increases in lipolytic enzymes, i.e., ATGL and β-HAD, in the Obese(-) group appear to prevent augmented deposition of lipid fractions compared with the subLean group, the development of metabolic syndrome reduced these compensatory cellular mechanisms and led to lipid accumulation (FFA, DAG and TAG) within the adipocytes. Furthermore, the cells differentiated from visADMSCs of both the obese groups had lower levels of the lipolytic enzymes than their counterparts obtained from subADMSCs, which indicates greater lipid turnover rate in the cells of subcutaneous provenance. Arner et al. reports that greater lipid turnover is observed in obesity when fat tissue is composed of numerous small adipocytes (hyperplasia) [24], and this was the case in our study. Nevertheless, most studies report that the progress of obesity is associated with a decreased lipid turnover in the cells [24]. In the present study, only the level of ATGL dropped with the advancement of obesity as evidenced in the cells originating from visADMSCs. The exact role of ATGL in metabolic abnormalities, however, is inconclusive since current literature reported either unchanged [39], increased [40] or decreased [41] levels of ATGL protein in obese humans or mice, with different patterns in male, female and mixed studied groups. Based on the data obtained, we may conclude that the adipocytes of the ADMSCs ancestry preserve several characteristics of mature fat cells in vivo. Particularly, the cells from the Obese(+) group (and to a lesser extent the Obese(-) group) reflect the phenotype observed in vivo in the early stages of obesity.

## 5. Conclusions

In summary, our study is the first to characterize in detail the phenotype of the adipocytes differentiated from the stem cells (ADMSCs) obtained from lean and obese patients (with and without metabolic syndrome). Particular emphasis was put on their broadly investigated lipid profile. We report several interesting findings regarding the differences in the examined parameters with respect to the tissue of origin and donor patients’ metabolic status. Several of the observed patterns reflect those that were previously reported in mature adipocytes in vivo. First of all, we observed that the cells of visADMSCs provenance were larger and displayed smaller granularity than their counterparts of subADMSCs ancestry. This points to their propensity to accumulate lipids in a few spacious vacuoles. Moreover, obesity led to further increases in the cell size and a drop in their granularity as evidenced by comparing the visObese(+) with visLean group. Secondly, we identified CD36/SR-B2 and FATP4 as the two most important FA transporters in the investigated adipocytes. Moreover, obesity appears to be associated with greater expression of the transporters (CD36/SR-B2 and FATP4) at the transcript (mRNA) and protein (total and cell surface) level. Overall, we found increased lipid concentration (TAG) in the cells of subcutaneous origin when compared with their counterparts from the other depot. Interestingly, the cells were also characterized by a boosted lipid synthesis (greater protein expression of FASN and DGAT1). Besides, obesity and metabolic syndrome were associated with further accumulation of TAG, DAG, and FFA in the cells that stemmed from subADMSCs accompanied by an increase in FASN and DGAT1 protein expression. Additionally, the cells of subcutaneous ancestry displayed greater lipid turnover than their visceral counterparts. Altogether, these data highlight that obesity significantly alters the phenotype of adipocytes differentiated from ADMSCs, and the progression of obesity towards metabolic syndrome further exacerbates these differences. These, differential responses of the adipocytes suggest that there is some memory effect of obesity influencing the functionality of progenitor adipocytes isolated from different fat depots.

## Figures and Tables

**Figure 1 cells-11-01435-f001:**
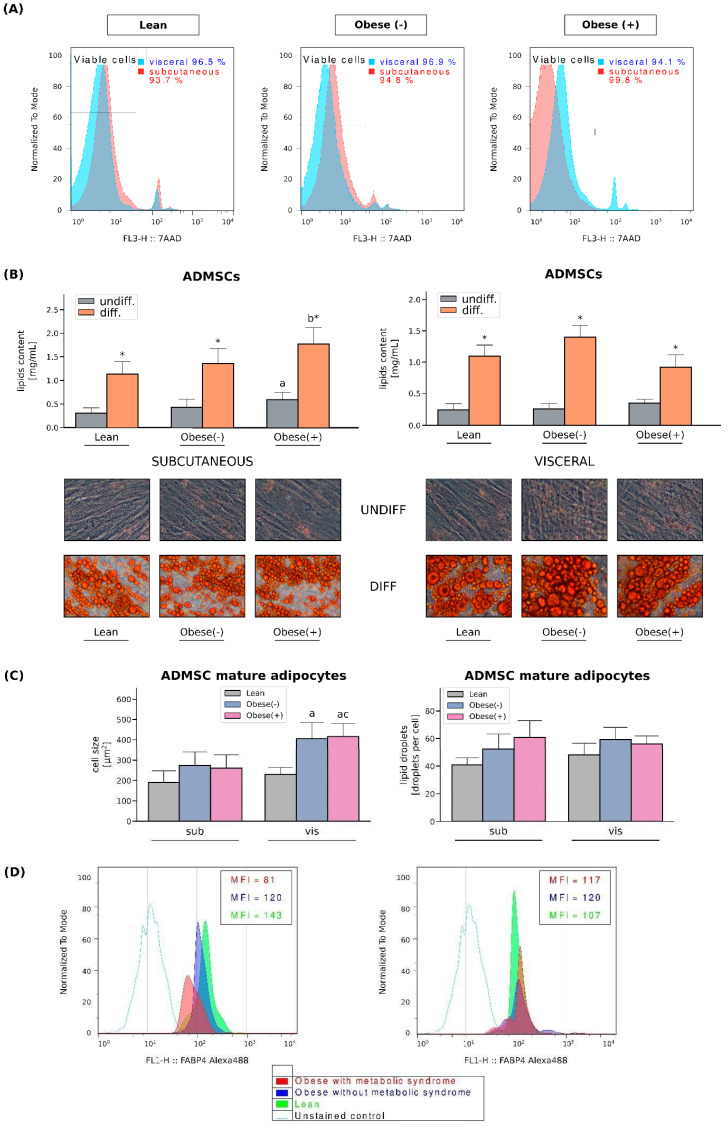
Viability of cultured ADMSCs and their differentiation into adipocytes. (**A**) Viability of ADMSCs derived from both subcutaneous and visceral adipose tissues. (**B**) Quantification of neutral lipids content in the undifferentiated and differentiated ADMSCs derived from lean, obese(-) and obese(+) individuals. Representative images of undifferentiated and adipogenically differentiated ADMSCs are shown. Lipid droplets inside the cytoplasm were stained with Oil Red O solution [200× magnification]. (**C**) Quantification of the average cell size [µm^2^] and the number of lipid droplets per cell. Bars and whiskers represent the mean and SD, respectively. Number of patients equals three (measurements taken in triplicate). a–difference vs. Lean group in the studied tissue; b–difference vs. Obese(-) group in the studied tissue; * significantly different from undifferentiated cells, *p* < 0.05. Designation of the groups: Obese(-)–obese without metabolic syndrome patients; Obese(+)–obese with metabolic syndrome patients. (**D**) Representative histograms of FABP4 expression in adipocytes differentiated form sub- and visADMSCs obtained from lean and obese subjects.

**Figure 2 cells-11-01435-f002:**
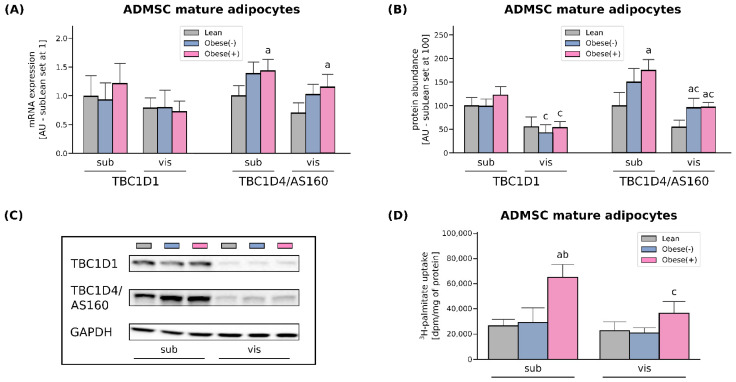
Gene (**A**) and protein expression (**B**) of AS160/TBC1D4 and its structural homolog TBC1D1 in adipocytes differentiated from ADMSCs derived from SAT and VAT of lean and morbidly obese women. Values are expressed in arbitrary units; the mean in adipocytes differentiated from the subADMSCs’ lean control was set at 1 or 100, respectively. (**C**) Representative Western Blot images are shown. (**D**) Measurement of ^3^H-palmitate uptake in fully differentiated adipocytes from subADMSCs and visADMSCs of lean and morbidly obese women. Values are expressed in DPM per mg of protein. a–difference vs. Lean group in the studied tissue; b–difference vs. Obese(-) group in the studied tissue; c–difference between adipocytes differentiated from visADMSCs vs. subADMSCs within the patient metabolic status. Data are presented as mean ± SD (*n* = 4 for each study group, measurements taken in duplicate for LCFA uptake and *n* = 3 for WB analysis). *p* < 0.05. Designation of the groups: Obese(-)–obese without metabolic syndrome patients; Obese(+)–obese with metabolic syndrome patients.

**Figure 3 cells-11-01435-f003:**
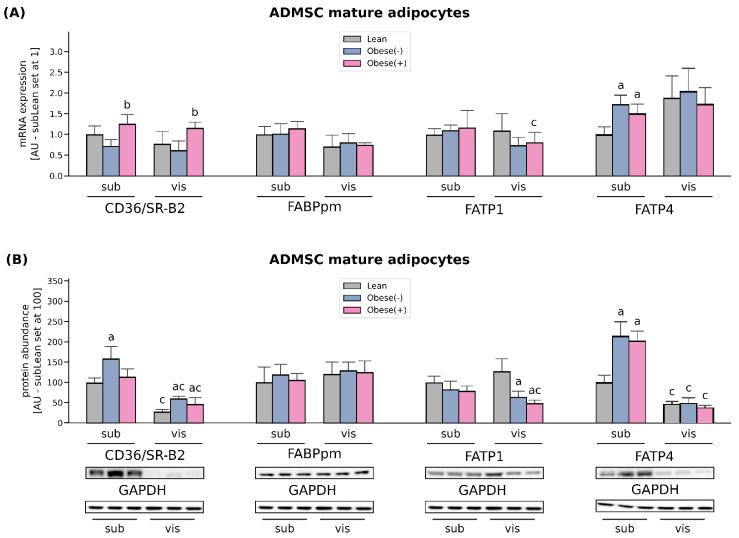
Quantification of (**A**) mRNA and (**B**) protein abundance of CD36/SR-B2, FABPpm, FATP1 and FATP4 in fully differentiated adipocytes derived from subADMSCs and visADMSCs of lean and morbidly obese women. Values are expressed in arbitrary units; the mean in adipocytes differentiated from the subADMSCs’ lean control was set at 1 or 100, respectively. Representative Western Blot images are shown. a–difference vs. Lean group in the studied tissue; b–difference vs. Obese(-) group in the studied tissue; c–difference between adipocytes differentiated from visADMSCs vs. subADMSCs within the patient metabolic status. Data are presented as mean ± SD (*n* = 4 for each study group, measurements taken in duplicate for rtPCR method and *n* = 3 for WB analysis). *p* < 0.05. Designation of the groups: Obese(-)–obese without metabolic syndrome patients; Obese(+)–obese with metabolic syndrome patients.

**Figure 4 cells-11-01435-f004:**
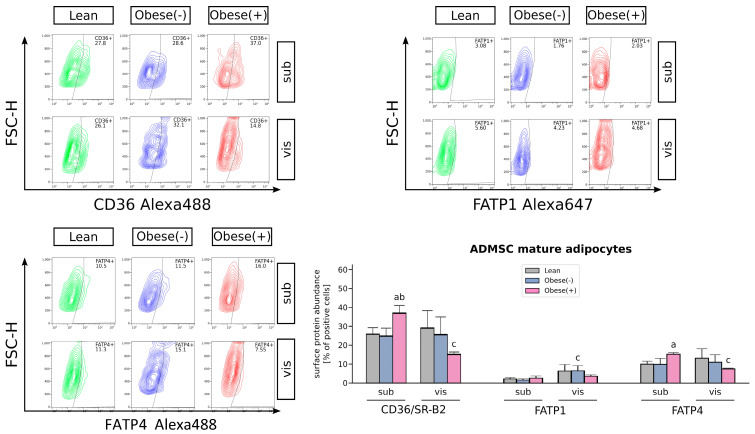
Flow cytometric analysis of cell surface expression of FA transporters in adipocytes derived from subADMSCs and visADMSCs of lean and morbidly obese women. The quantification of the cell populations staining positively for CD36/SR-B2, FATP1 and FATP4. Representative plots are shown. a–difference vs. Lean group in the studied tissue; b–difference vs. Obese(-) group in the studied tissue; c–difference between adipocytes differentiated from visADMSCs vs. subADMSCs within the patient metabolic status. Data are presented as mean ± SD (*n* = 3 for each study group, measurements taken in duplicate). *p* < 0.05. Designation of the groups: Obese(-)–obese without metabolic syndrome patients; Obese(+)–obese with metabolic syndrome patients.

**Figure 5 cells-11-01435-f005:**
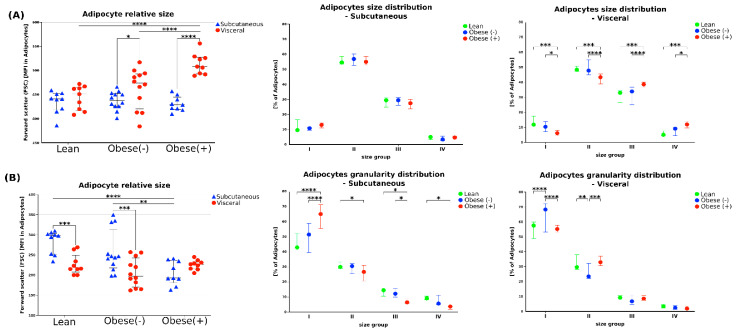
The size and the granularity of adipocytes differentiated from subADMSCs and visADMSCs of lean and morbidly obese women. The cells were characterized based on the forward scatter (FSC, relative to adipocytes size) and side scatter (SSC, relative to adipocytes internal structure) during flow cytometry analysis. (**A**) Adipocytes relative size and size distribution. (**B**) Adipocytes relative granularity and granularity distribution. Numbers indicate ranges used to classify the cells to different groups based on the cell size (I: 0–275, II: 275–550, III: 550–825, IV: 825–1100) or granularity (I: 0–250, II: 250–500, III: 500–750, IV: 750–1100). *—*p* < 0.05; **—*p* < 0.01; ***—*p* < 0.001; ****—*p* < 0.0001. The inner horizontal line represents the median. Whiskers: 25–75 percentile. *n* = 3 for each study group, measurements taken in triplicate. Designation of the groups: Obese(-)–obese without metabolic syndrome patients; Obese(+)–obese with metabolic syndrome patients.

**Figure 6 cells-11-01435-f006:**
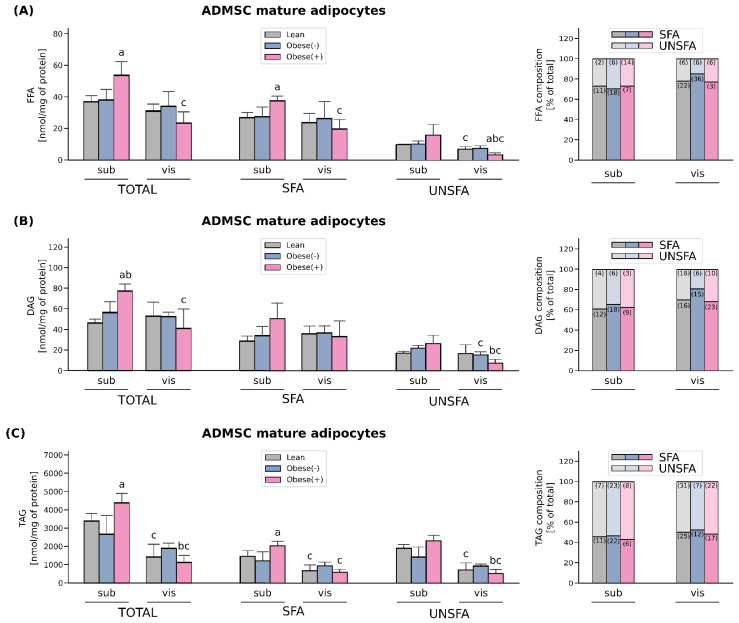
Lipid content and FA-profile in differentiated adipocytes derived from subADMSCs and visADMSCs of lean and morbidly obese women. The amount of total saturated fatty acid species (SFA), unsaturated fatty acid species (UNSFA) and FA distribution in (**A**) FFA, (**B**) DAG, (**C**) TAG. Values are expressed in nmol per mg of protein. a–difference vs. Lean group in the studied tissue; b–difference vs. Obese(-) group in the studied tissue; c–difference between adipocytes differentiated from visADMSCs vs. subADMSCs within the patient metabolic status. Bars and whiskers represent the mean and SD (*n* = 4 for each study group, measurements taken in triplicate). For the percentage fatty acid composition, numbers in brackets inside bars represent SD. *p* < 0.05. Designation of the groups: Obese(-)–obese without metabolic syndrome patients; Obese(+)–obese with metabolic syndrome patients.

**Figure 7 cells-11-01435-f007:**
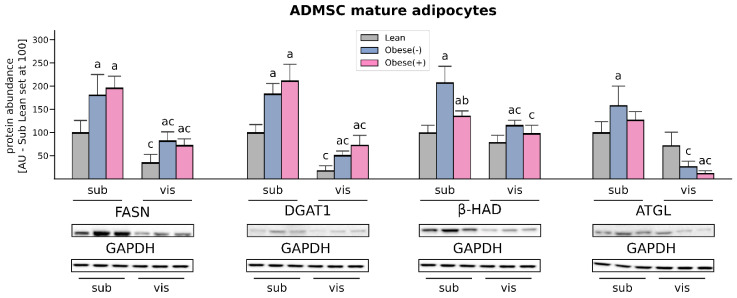
Protein content of enzymes involved in fatty acid synthesis and utilization (FASN, DGAT1, β-HAD, ATGL) in adipocytes differentiated from subADMSCs and visADMSCs of lean and morbidly obese women. Values are expressed in arbitrary units; the mean in adipocytes differentiated from the subADMSCs’ lean control was set at 100. Representative Western Blot images are shown. a–difference vs. Lean group in the studied tissue; b–difference vs. Obese(-) group in the studied tissue; c–difference between adipocytes differentiated from visADMSCs vs. subADMSCs within the patient metabolic status. Data are presented as mean ± SD (*n* = 3 for each study group, measurements taken in duplicate). *p* < 0.05. Designation of the groups: Obese(-)–obese without metabolic syndrome patients; Obese(+)–obese with metabolic syndrome patients.

**Figure 8 cells-11-01435-f008:**
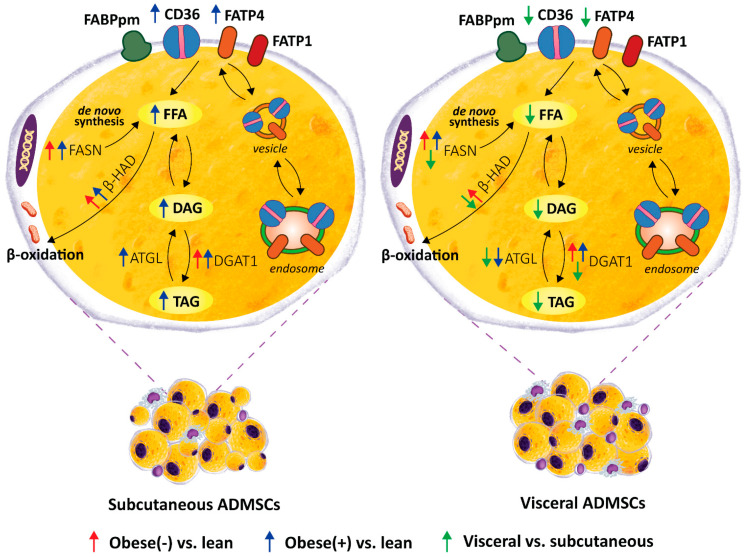
Graphical summary of the lipid profile of mature adipocytes derived from ADMSCs of lean and obese (with/without metabolic syndrome) postmenopausal women. A depot-specific pattern, with more pronounced changes present in the adipocytes obtained from subADMSCs was observed. Namely, morbid obesity triggered a relocation of fatty acid transporters i.e., CD36/SR-B2 and FATP4, from intracellular vesicles to the cell surface, which caused an increased LCFA influx (^3^H-palmitate uptake) and promoted the accumulation of lipids in these cells. This was accompanied by a greater lipid turnover rate in the cells of subcutaneous provenance (greater protein expression of lipolytic enzymes, i.e., ATGL and β-HAD). On the other hand, the adipocytes of visADMSCs origin had lower lipid content and expression of lipid metabolizing enzymes, i.e., ATGL, β-HAD and FASN, when compared to their counterparts of subADMSCs ancestry.

## Data Availability

The data presented in this study are available on request.

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
