# Peer review of "The Phenotype of the Adipocytes Derived from Subcutaneous and Visceral ADMSCs Is Altered When They Originate from Morbidly Obese Women: Is There a Memory Effect?"

_cells, 2022, doi:10.3390/cells11091435_

Round 1

Reviewer 1 Report

The Authors address an interesting topic about the phenotypic and metabolic characteristics, particularly lipid profile, of fully differentiated adipocytes derived from adipose mesenchymal stem cells (ADMSCs) of lean and obese (with/without metabolic syndrome) postmenopausal women. Goal of the study is to highlight that obesity significantly alters the phenotype of adipocytes differentiated from ADMSCs, and its obesity progression towards metabolic syndrome further exacerbates these differences suggesting that there is a memory effect of obesity on influencing the functionality of progenitor adipocytes isolated from different fat depots.

The manuscript is generally well written in a clear and detailed manner.

I believe the following points need to be addressed:

  • In the first paragraph of Results related to the characterization of human ADMSCs, it should be better to add the data obtained from the cell viability assay determined via Trypan Blue as well as representative images ADMSCs cell morphology for each experimental condition.
  • As regards ORO staining for the identification of lipid droplets, the representative micrographs should be associated to an evaluation of the average size and number of lipid droplets/cell for each experimental condition.
  • In Figure 1A related to the quantification of neutral lipids in visADMSCs, the lipids content of differentiated visADMSCs derived from obese (+) individuals is comparable to the lean ones, unlike those of obese (-). How could the Authors explain this?
  • All the original immunoblots attached to the manuscript show multiple bands and sometimes the whole protein profile separated electrophoretically by SDS-PAGE. It seems that the antibodies are not really specific and that the Authors choose the immunoreactive bands just relying on the size. The Authors should improve the quality of immunoblots. In addition, sometimes the intensity of the immunoreactive bands is really too low to carry out the densitometric analysis [such as for example TBC1D1 and TBC1D4/AS160 in visADMSCs (Fig. 2C) or CD36/SR-B2 in visADMSCs (Fig. 3B)].
  • All the values expressed as a percentage should be accompanied by the statistics (SEM or SD).
  • The Discussion section is too long and articulated, the Authors should simplify and shorten it. I would suggest to insert in this section a working model that graphically summarizes all the results.

Sincerely

Reviewer 2 Report

1.My major concern is about the used statistical analysis tests. I expected that comparisons between more than two groups be conducted using ANOVA or it's non-parmetric equivalent Kruskalwalis  followed by appropriate post tests.

Minor concerns:

1. Please add details for the qPCR, including temperature, number of cycle, etc in the method section.

2. Please include annealing temperature and ampliqon size in Table S1.

Reviewer 3 Report

Manuscript titled “The phenotype of the adipocytes derived from subcutaneous and visceral ADMSCs is altered when they originate from morbidly obese women. Is there a memory effect?” by Miklosz et al. investigated the phenotypic and metabolic characteristics of primary differentiated subcutaneous and visceral adipocytes from lean and obese (with and without metabolic syndrome) derived from postmenopausal women. By using these adipocytes, and PCR and protein analysis, flow cytometry and gas liquid chromatography the authors describe how the obese subcutaneous derived adipocytes have a higher gene and protein expression of TBC1D4; CD36/SR-B2; FATP4 along with higher palmitate uptake, higher FFA, DAG and TAG content compared to their lean counterparts and compared to obese visceral adipocytes.

The subject being investigated is of great significance. The paper was nicely written and overall, the data is well presented. However, some aspects of the study that should be addressed.

  1. The authors have good controls in the study (lean vs obese; obese vs obese with metabolic syndrome). They state that adipocytes are from 4 women for each group (page 3 of 20, line 105-110). However, it is not clear if the adipocytes were pooled, or they had 4 separate biological replicates. Please describe more precisely this aspect since 4 biological replicates is different from technical replicates.
  2. Regarding the flow cytometry: how many cells were chosen? How many cells were recovered?
  3. Please add the catalog number of the antibodies used, as well as the dilutions of the primary antibodies used (Line 203 of page 5 of 20).
  4. Line 270 on page 6 of 20: “Number of patients equals three…” you state that you had 4 patients for each group. Why was one left out from this analysis?
  5. The full blots provided by the authors show replicates of 3 (technical or biological?). From the full blot a section is used in the main paper. However, the figure legend states that data are presented as mean +/- SD (n=4 for each study group, measurements taken in duplicate): Line 305 of page 7 of 20. Please clarify.
  6. Page 7 of 20, Figure 2: all data are showing the expression of TBC1D1 and TBC1D4 at basal sate. What happens to TBC1D1 and TBC1D4/AS160 when the differentiated adipocytes are stimulated? For example, treatment with insulin.
  7. Along with the impaired palmitate uptake, it would really nice to show if there is any impairment in lipolysis since obese patients have impaired basal and stimulated lipolysis due to catecholamine resistance.
  8. Point 7 goes along with data presented in Figure 7 where the expression of ATGL seems to be increased in obese subcutaneous fat, but decreased in obese visceral fat.
  9. These experiments might help when you discuss that obese subcutaneous and obese visceral adipocytes handle fatty acid differently (Line 535, page 15 of 20).

Round 2

Reviewer 2 Report

Dear authors

Thank you for addressing my concerns.

Reviewer 3 Report

Dear Authors,

thank you for addressing my comments. You have added to the main text the details that were suggested. Below are a few additional comments:

1) Please add to the main text the number of cells recovered for your flow analysis.
